# Bicyclobutanes as unusual building blocks for complexity generation in organic synthesis

Maxim Golfmann [ID] [1] & Johannes C. L. Walker [ID] [1✉]

Bicyclobutanes are among the most highly strained isolable organic compounds and their associated low activation barriers to reactivity make them intriguing building-blocks in organic chemistry. In recent years, numerous creative synthetic strategies exploiting their heightened reactivity have been presented and these discoveries have often gone hand-in-hand with the development of more practical routes for their synthesis. Their proclivity as strain-release reagents through their weak central C–C bond has been harnessed in a variety of addition, rearrangement and insertion reactions, providing rapid access to a rich tapestry of complex molecular scaffolds. This review will provide an overview of the different options available for bicyclobutane synthesis, the main classes of compounds that can be prepared from bicyclobutanes, and the associated modes of reactivity used.

Bicyclobutane (**1**) is the smallest fused hydrocarbon and has intrigued generations of chemists for over 100 years[1]. Its molecular structure, with bond angles and orbital hybridisations far from those usually encountered in unstrained hydrocarbon molecules, provided the initial challenge—how to prepare such a highly distorted and strained structure? Once useful synthetic routes were established, the unusual reactivity of the bicyclobutane (BCB) structure could be investigated and has inspired numerous creative synthetic methodologies. In this review, we will provide a summary of the available synthetic approaches to BCBs and the main types of reactivity of the BCB scaffold from the perspective of their application in the synthesis of complex molecular structures. This task is one of increasing importance to the organic chemistry community; there is renewed demand for practical routes to more structurally diverse compounds from the pharmaceutical industry as it searches for the next generation of therapeutics[2–5]. The early chemistry of BCBs has been reviewed previously[6–8], and the most recent reviews[9,10] focus mainly on the strain-release reactions to cyclobutane derivatives and carbene insertion reactions with BCBs[11]. In this review, we approach the chemistry from the perspective of the different structural classes of compounds that can be prepared using BCBs, and include the most recent developments in this rapidly expanding field.

## Structure of bicyclobutanes

Bicyclobutane (**1**) adopts a so-called "butterfly" conformation, with the two cyclopropane rings distorting from a planar conformation by 120–125° (Fig. 1A)[12]. Interestingly, all C–C bonds also have approximately the same length of 1.50 Å[12]. For BCB derivatives, the bond lengths and distortion angle are dependent on the substitution of the BCB scaffold, with axial bridge substituents forcing an opening of the butterfly structure, an increase in the angle between the cyclopropane rings, and a lengthening of the central C–C bond[13]. The internal cyclopropane angles are around 58–62°, a significantly deviation from the typical 109 °C–C bond angles found in saturated hydrocarbons, and contribute to the well-known and substantial strain energy of BCBs (≈64 kcal mol⁻¹, more than double the ≈28 kcal mol⁻¹ strain energy of an individual cyclopropane)[14]. The marked structural distortions lead to the unusual properties of BCBs,

[1] Institut für Organische und Biomolekulare Chemie, Georg-August-Universität Göttingen, Tammannstr. 2, 37077 Göttingen, Germany.
✉email: johannes.walker@chemie.uni-goettingen.de

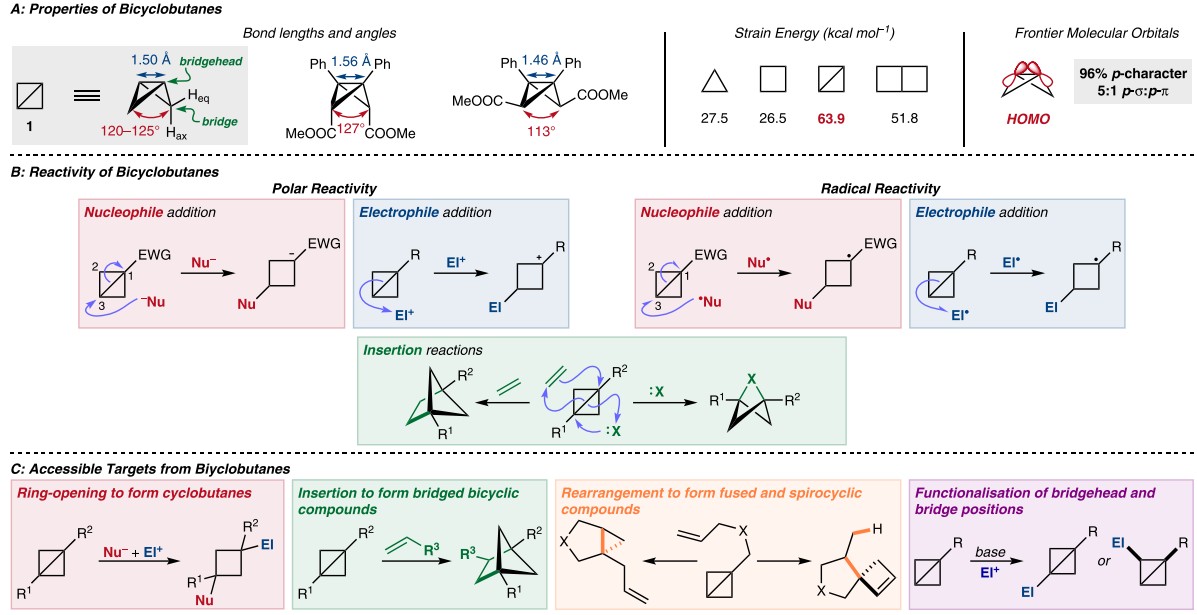

**Fig. 1 Introduction to bicyclobutanes. A** Properties of bicyclobutanes including bond lengths and angles, the high strain energy of bicyclobutanes in comparison to related structures, and the nature of the highest occupied molecular orbital of bicyclobutanes. **B** The most important general reactivities of the bicyclobutane scaffold: the addition of polar and radical nucleophiles and electrophiles and the insertion of various one- and two-atom fragments to bridged systems. **C** The structural classes of compounds that can be prepared from bicyclobutanes. aq axial, eq equatorial, HOMO highest occupied molecular orbital, EWG electron-withdrawing group, Nu nucleophile, El electrophile.

including their ability to ring-flip between different configurations at high temperature. Woodward and Dalrymple experimentally observed the interconversion of exo–exo to endo–endo isomers of doubly bridge-substituted BCBs at 120 °C[15]. The frontier molecular orbitals of BCBs, especially those relating to the central C–C bond, deserve particular attention. An early qualitative Walsh model described by Pomerantz and Abrahamson proposed significant π-character for the central C–C bond (the HOMO), and its construction from unhybridized 2p orbitals[16]. Later MO-SCF calculations by Newton and Schulman also suggested strong p-character, reporting that the central bond is $sp^{24.3}$ hybridised, or 96% p in character[17]. However, calculations by Schulman and Fisanick indicated that the molecular orbital has only limited π-character, being made up of approximately 5:1 p-σ:p-π contributions[18]. The authors state that "the central bond is best described as arising from largely σ interaction of two unhybridized p orbitals." Pomerantz and Hildebrand also used $^1J_{C–C}$ coupling constants in 1-cyanobicyclo[1.1.0]butane to calculate an s-contribution to the C–C bond of 10.8%[19]. In support of the proposed π-character, conjugation between two bridgehead phenyl groups through this C–C bond has been observed by UV-Spectroscopy[15]. Whitman and Chiang later concluded that any classical description of the C–C bond is "inadequate," with their calculations indicating that the bond is made up mainly of two p-orbitals at ~45° to the central axis[20]. While this bears some resemblance to a conventional π-bond, the electronic distribution is concentrated in one direction above axis, rather than the expected symmetric distribution. The bridgehead C–H bond also accounts for some of the unusual reactivity of BCBs; it is strongly polarised and can be deprotonated with strong organometallic bases[20]. At this time, theoretical investigations into the characteristics of the LUMO are extremely rare[21].

## Reactivity of bicyclobutanes
The reactivity of the BCB framework is dominated by the strained central C–C bond, and reactions that lead to its cleavage and the release of strain energy are highly favoured (Fig. 1B). This has

been broadly exploited, including in both polar nucleophilic and electrophilic addition, and insertion-type reactions. Reactions with nucleophilic and electrophilic radicals have also been reported. When an electron-withdrawing group is present at the bridgehead position, conjugate addition-type reactivity that is reminiscent of α,β-unsaturated systems leads to cleavage of the C–C bond and addition of a nucleophile. The HOMO of the BCB, mostly localised on the central C–C bond, can participate in electrophilic addition reactions; here, electrophile addition occurs to reveal the most substituted carbocation, which is then attacked by a nucleophile. This use of a single C–C bond of predominantly σ-character in electrophilic addition reactions is rather unusual. One other important class of BCB reactivity is insertion-type reactions; certain π-systems and carbenes can react with the central C–C bond to furnish bridged bicyclic motifs. Despite the theoretical investigations discussed earlier which suggest a predominantly σ-character to the C–C bond, this reactivity is more reminiscent of π-bonds and concerted cycloaddition mechanisms. Stepwise mechanisms are nonetheless often proposed for these examples.

## Accessible targets from bicyclobutanes
The diverse reactivity of bicyclobutanes allows them to be flexible building blocks towards a range of structural targets, from relatively simple cyclobutanes to more complex bridged, fused, and spirocyclic systems. Derivatisation reactions which leave the highly strained core intact are also possible, and beyond providing more densely functionalised bicyclobutanes which may be of interest on their own, these reactions also provide more elaborate building blocks for the other reactivities discussed. All of these possibilities will be discussed in more detail later.

## Synthesis of bicyclobutanes
Early attempts at the synthesis of a BCB were plagued by unfortunate mischaracterisations. Both Perkin and Simonsen[1] and Guthzeit and Hartmann[22] thought they had succeeded, but follow-up investigations concluded that BCB structures were not

obtained[23,24]. The breakthrough occurred in 1959, when Wiberg and Ciula were finally able to prepare and characterise BCB **5** (Fig. 2)[25]. Starting from epoxide **2**, ring-opening to a dibromide which was used in the dialkylation of diethylmalonate led to cyclobutane **3**. Decarboxylation and exchange of the benzyl ether functionality for a bromide provided **4**, which was ideally set up for ring-closure to give the much sought-after BCB framework. The $^1H$ NMR spectrum revealed three distinct aliphatic C–H environments (corresponding to $H_{ax}$, $H_{eq}$, and $H_b$) and no alkenyl C–H signals (which would have been expected in any competing elimination reaction), and was consistent with the proposed BCB structure. The first X-Ray crystal structure of a bicyclobutane was obtained by Johnson and Schaefer who successfully crystallised 1,3-dicyanobicyclobutane[26].

Much work since the pioneering synthesis of BCB **5** by Wiberg and Ciula has been devoted to establishing more efficient methods for BCB synthesis. Today, many routes are known (Fig. 2). The most commonly used can be categorised by the manner of cyclisation; either through formation of the internal C–C bond (here termed transannular cyclisation, Path A) or one of the peripheral C–C bonds (here termed side-chain cyclisation, Paths B and C). Another less used approach involves forming the cyclopropane rings by cyclopropanation (Path D). Importantly, however, this latter approach provides the only known route to prepare enantioenriched BCBs. In general, most of the routes described lead to monosubstituted BCBs bearing one electron-withdrawing substituent at the bridgehead position (typically a nitrile, sulfone or carbonyl). Routes to disubstituted BCBs with a second bridgehead substituent are less common and de novo routes to BCBs with bridge substitution are rarer still.

The transannular cyclisation approach (Path A) is based on the seminal work of Wiberg and Ciula and can be used to prepare both mono- and disubstituted BCBs. There are three general routes that use this strategy. Reaction of 3-methylenecyclobutanecarbonitrile (**6**) with a halogen acid (HCl, HBr, or HI) installs a halogen leaving group at the 3-position of the cyclobutane, which facilitates BCB formation on addition of base. This was first reported by Blanchard and Cairncross who used HI to generate nitrile-substituted BCB **8**[27]. HCl[28] and HBr[29] have also been used since. Wipf and co-workers recently used this method to prepare multi-gram quantities of BCB **8**, using NaH to effect cyclisation from bromide **7**[30]. Large quantities of BCBs have also been prepared starting from 1,1-cyclobutanedi-carboxylic acid (**9**). This approach was developed by Hall and co-workers who built on the remote chlorination of diacid **9** by Lampman and Aumiller[31] to prepare 3-chlorinated cyclobutanes. These could be cyclised under basic conditions to give ester-substituted BCBs **11**[32]. Anderson and co-workers later used this method to prepare multi-gram quantities of the intermediates **10** en-route to amide-substituted BCB **12**[33,34]. The use of 3-oxocyclobutanecarboxylic acid (**13**) for BCB synthesis is a comparatively recent invention[35]. Addition of an appropriate nucleophile to the carbonyl functionality enables derivatisation of the 3-position and the formed tertiary alcohol can be easily converted to an appropriate leaving group for cyclisation. Ma[36] and Mykhailiuk[37] independently demonstrated this approach to prepare phenyl-substituted BCB **15**. Alternative nucleophiles have been used including vinyl magnesium bromide (to give **16**)[37,38], sodium borohydride (to give **17**)[36] and an in situ generated trifluoromethyl anion (to give **18**)[39]. This approach can provide access to different bridgehead disubstituted BCBs, but the incorporation of the second substituent in the first step and three subsequent transformations makes it a relatively time-consuming approach. Tilley and co-workers reported a related transannular cyclisation approach, using an unusual 1,3 γ-silyl elimination to form the BCB (not shown)[40].

Side-chain cyclisation is arguably the most well-established route to BCBs, and is commonly (but not exclusively) used to prepare monosubstituted BCBs with a single-electron-withdrawing substituent at the bridgehead position. This approach was pioneered by Gaoni, who used epoxysulfone **19** as the starting point for his synthesis of sulfone-substituted BCB **21** (Path B)[41–43]. Deprotonation leads to intramolecular epoxide ring-opening and formation of the first cyclopropane ring. Activation of the formed alcohol as mesylate **20** then enables a second ring-closure to afford sulfone-substituted BCB **21**. The excellent electrophilicity of the sulfone-substituted BCBs (vide infra) has led to this route being adopted by numerous research groups, and a range of electron-rich and electron-poor aryl sulfone BCBs including **22–24** have been prepared[33,44–48]. This route suffers somewhat from the required preparation of the epoxysulfone starting material **19**, but one elegant solution to this was recently published by Jung and Lindsay, who reported a one-pot procedure to BCB **21** starting from methyl phenyl sulfone (**25**) and epichlorohydrin[49]. They were also able to prepare sulfonamides such as **27** and bridge-substituted BCBs such as **28** using this approach.

A related and also much used route to BCBs exploits the cyclopropanation of alkenes with dibromocarbene (Path C). The first example of this was reported by Scattebøl, Baird, and co-workers who prepared disubstituted bromide **41**[50]. Subsequently, the route has most commonly been used to prepare mono-substituted BCBs. Bromide **31** was synthesised by Düker and Szeimies, who started by reacting allyl chloride (**29**) with dibromocarbene, generated under basic conditions from bromoform[51]. The isolated dibromocyclopropane **30** was treated with MeLi to effect Li–Br exchange and enable ring-closure to give bromide-substituted BCB **31**. The strategic advantage of this approach is that bromide **31** can be lithiated to form Li-BCB **32**, which may be used to attack a range of different electrophiles. Sulfinate esters (to sulfoxide **33**)[52–54], boronate esters (to boronate **34**)[55], chloroformates (to ester **5**)[56], acyl nitriles (to amide **35**)[57], Weinreb amides (to ketone **36**)[57], and aldehydes or imines (to alcohol **37** or amine **38**)[58] have all been used as electrophiles in this reaction. Of particular interest here is the preparation of sulfoxide **33**. Aggarwal and co-workers noted that this species is crystalline, easily isolable, and can be stored until required[52,53]. The analogous sulfoxide–lithium exchange (using tBuLi) is able to regenerate Li-BCB **32**, which can then be used as before. The dibromocyclopropane approach has also been sporadically used to prepare polysubstituted BCBs. In addition to **41**, Scattebøl, Baird, and co-workers also prepared bridge-disubstituted BCB **40**[50], and Wipf and co-workers used dimethyl malononitrile as an electrophile to furnish aryl-substituted nitrile **39**[30]. A small number of other examples are also known[51,59]. The potentially lengthy preparation of the required substituted allyl chlorides is likely to be a limiting factor to the more widespread adoption of this approach to polysubstituted BCBs.

A less used but conceptually distinct route to BCBs is through cyclopropanation (Path D)[60]. Fox[61] and Davies[62] both devised Rhodium-catalysed asymmetric intramolecular cyclopropanations, with the former transforming α-diazo ester **42** into BCB (−)−**44** in 95% ee. Double cyclopropanations from alkynes to racemic BCBs have been pursued by Mahler[63], Doering and Coburn[64], and Wipf[65]. Through directed evolution, Arnold and co-workers were able to develop an enzyme capable of effecting asymmetric cyclopropanation and could use this to prepare ester-substituted BCB (−)-**43** in >98% ee[66]. This example also provides access to BCBs with substitution on both methylene bridges. These examples are notable for being the only known de novo routes to enantioenriched BCBs.

A limited number of other routes are also known that do not fall under the above categories but are nevertheless conceptually interesting and are included for completeness. A reductive

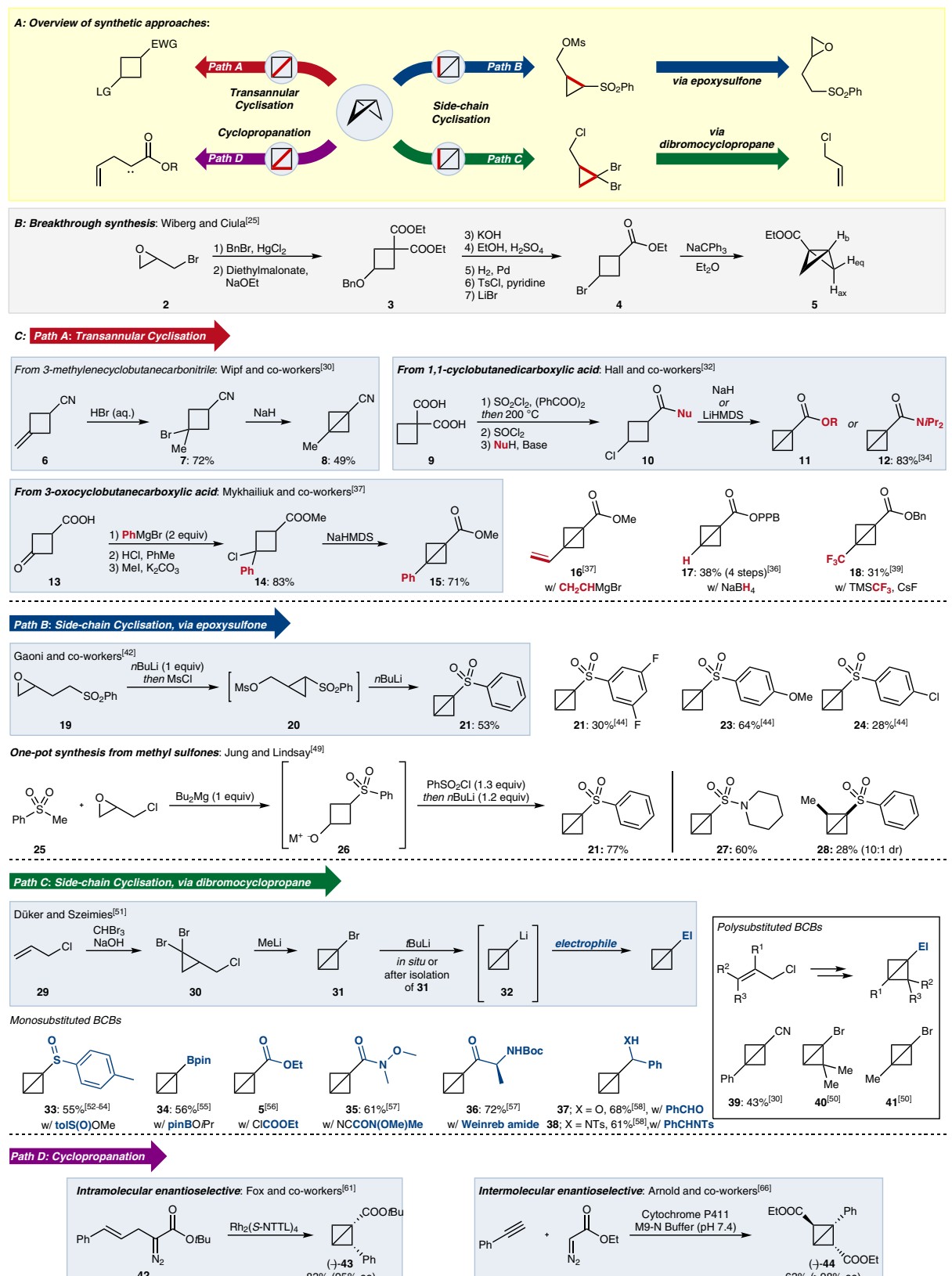

**Fig. 2 Overview of synthetic approaches of BCBs. A** Overview of the different synthetic approaches to bicyclobutanes categorised by the bonds formed. **B** The landmark first synthesis of a bicyclobutane derivative by Wiberg and Ciula. **C** The most commonly used synthetic routes to bicyclobutanes, highlighting the derivatives that can be formed with each route with a focus on the substitution pattern and functional groups that can be incorporated. HMDS bis(trimethylsilyl)amide, TMS trimethylsilyl, Ms methane sulfonyl, tol tolyl, pin pinacol, Ts *para*-toluene sulfonyl, NTTL *N*-1,8-naphthaloyl-(*S*)-*tert*-leucine; Boc *tert*-butoxycarbonyl, EWG electron-withdrawing group, LG leaving group, Nu nucleophile, El electrophile.

cyclisation of dihalocyclobutanes has also been disclosed for the synthesis of bicyclobutane (**1**) itself[67], and Chang and Dougherty published a synthesis of bicyclobutane (**1**) via $N_2$ extrusion[68].

## Substituted cyclobutanes by strain-release ring-opening

As previously mentioned (see Fig. 1C), the reactivity of BCBs is dominated by the highly strained central C–C bond. The high ring strain of BCBs has most commonly been exploited in the context of using BCBs as "spring-loaded" entities for cyclobutane synthesis (Fig. 3). In this setting, the nucleophile adds to the distal position of BCBs bearing electron-withdrawing groups in a strain-release ring-opening reaction and the central weak C–C bond is broken (Fig. 3A). Disubstituted cyclobutanes are most commonly formed, but tri- and even tetrasubstituted products may also be obtained. Studies by Gaoni and co-workers showed that polar nucleophiles attack along the *endo* trajectory[69] and the formed anionic species can then be protonated or reacted with additional electrophiles. Under conditions where fast protonation is possible (strong acid or protic solvent) the anti-diastereomer of the cyclobutane is formed[70], otherwise the electron-withdrawing substituents takes up a pseudo-equatorial position and the syn-diastereomer forms on protonation. These reactions can be categorised by the type of nucleophile used and the reaction mechanism. Reactions using 2-electron-based nucleophiles are particularly well studied (here termed polar strain-release) with radical-based nucleophiles recently becoming more popular (here termed radical strain-release). The rearrangement chemistry of boronate-substituted BCBs and Umpolung BCB activation with Cobalt catalysts further highlight the intriguing synthetic possibilities available when harnessing the "spring-loaded" nature of BCBs.

Early work in strain-release ring-opening reactivity was performed by Gaoni and co-workers, who showed that strong nucleophiles like organocuprates[71] and metal hydrides (e.g. $LiAlH_4$)[72] could add to sulfonyl-substituted BCBs. The anionic intermediates could then be protonated or reacted with other electrophiles. Methyl substituted BCBs such as **45** were also studied, with the selective formation of *trans*-dimethyl-substituted cyclobutane **46** providing evidence for the endo attack of nucleophiles[69].

As part of their Rhodium-catalysed asymmetric intramolecular cyclopropanation reactions (see Fig. 2), Fox and co-workers described a one-pot protocol for cyclopropanation/organocuprate conjugate addition/electrophile addition[61]. Starting from α-diazo ester **47**, cyclopropanation led to BCB **48** which was reacted in situ with an organocuprate nucleophile. Protonation of the resulting enolate or electrophile addition provided a library of tri- or tetrasubstituted enantioenriched cyclobutanes including **50–53**. Alkyl nucleophiles (to **50–52**) or aryl nucleophiles (to **53**) could be used and allyl (to **51**), thiol (to **52**), and acyl electrophiles (to **53**) could all be successfully employed to afford diverse tetrasubstituted BCBs. In these cases, the opposite diastereomer to that expected from the work of Gaoni and co-workers was obtained[69]. The use of HCl as a proton source resulted in very low diastereoselectivity (1.3:1, major diasteromer shown) in the formation of **50**, but the bulkier proton source BHT led to increased and opposite selectivity (1:6 dr). Epimerisation with KOtBu led to isolation of **50** in 1:19 dr. The authors later applied their methodology to the total synthesis of Piperarborenine B[73].

One of the best studied areas of BCB strain-release chemistry is the addition of amine derivatives to BCBs to give cyclobutyl amines. Pioneered by Baran and co-workers, the addition of free amines to arylsulfonyl-substituted BCBs proceeds at room temperature and can also be extended to other "spring-loaded" systems including propellanes, 1-azabicyclobutanes, and

housanes[44,45]. The identity of the aryl sulfone group was critical, with electron-deficient aryl groups, particularly 3,5-difluorophenyl groups (as in **22**), proving most efficient. The arylsulfonyl group can be removed under reductive conditions (Conditions **A**) or further derivatised (Conditions **B**) to give a flexible set of reactions to substituted aminocyclobutanes. The scope of this reactivity is broad, with acyclic amines (to **54**) and cyclic amines (to **55**) both tolerated and heterocycle-substituted amines (to **56**) also reacting successfully. This strategy could also be applied to the cyclobutylation of natural product-like amines in the context of late stage modification (as in **57**). Electrophiles including allyl bromide (to **58**), NFSI (to **59**), and deuterated methanol (to **60**) provide ready access to cyclobutanes with useful functional group handles or pharmaceutically relevant substituents.

Other heteroatom-based nucleophiles have also been reacted with BCBs. Gaoni and co-workers employed TMGA (tetramethylguanidinium azide) to add azides to BCBs including **61** to give cyclobutane **62**. Further derivatisations of these products yielded compounds including unnatural cyclobutane-based α-amino acids (not shown)[74,75]. Malins and co-workers utilised thiols as nucleophiles and further investigated the utility of this approach for peptide labelling. Besides conventional thiols including *para*-methoxy benzyl thiol (to **63**) the thiolation of BCBs by cysteine residues proved to be successful[57]. The use of BCBs in bioconjugation chemistry is a growing area of interest, with their use in protein labelling[76] or as [131]I-radiolabels both being reported[77]. Phosphines in the form of boranophosphanes were harnessed as nucleophiles by Wipf and co-workers to give cyclobutyl phosphines such as **64**[30]. *H*-Phosphonates could also be used as nucleophiles under adapted reaction conditions and the products could be combined with BINOL derivatives to give bidentate phosphine–phosphite ligands for Rh(I)-catalysed asymmetric hydrogenations.

Beyond the polar additions discussed above, the addition of single-electron radical nucleophiles to BCBs has recently begun to be investigated (Fig. 3B). Their high strain energy makes BCBs excellent radical acceptors. The radicals used are typically prepared in situ by thermal, photochemical, or redox activation. Lin and co-workers reported a Ti(III)-catalysed Cl atom abstraction from tertiary alkyl chlorides to provide tertiary alkyl radicals[46]. These radicals could be added to a variety of α,β-unsaturated carbonyl systems as well as BCB **21**, leading to the formation of alkyl-substituted cyclobutane **65**. Photoredox catalysis has emerged as a powerful tool for the generation of radicals under mild conditions[78]. Jui and co-workers harnessed iridium photoredox catalysts to generate α-amino radicals from *N*-methyl anilines and demonstrated their addition to BCBs[79]. Alkyl amine-substituted cyclobutanes including **66** could be prepared in this way. Another prominent class of photoredox-activated radical precursors are carboxylic acids. Cintrat and co-workers developed an iridium-catalysed decarboxylation reaction to generate α-amino or α-oxy stabilised radicals for addition into BCBs to give cyclobutanes **67**[47]. The reaction tolerates carboxylic acids with various ring sizes, cyclic or acyclic ethers and protected amines (PG = Boc (*tert*-butoxycarbonyl), Bac (*tert*-butylaminocarbonyl), Piv (pivaloyl), Cbz (carboxybenzyl)), and even drugs and peptides as nucleophiles in the context of late stage functionalisation. Finally, Studer and co-workers developed a BCB hydrosilylation reaction to give cyclobutyl silanes **68**[80]. A photoredox-catalysed oxidation of disilanes generated a silyl radical, which could add to BCBs as well as electron-deficient alkenes.

A distinct class of BCB activation by strain-release has been investigated by Aggarwal and co-workers, who developed a suite of boronate rearrangements from common intermediate BCB-boronate **69** (Fig. 3C). In these reactions, the chemistry of the

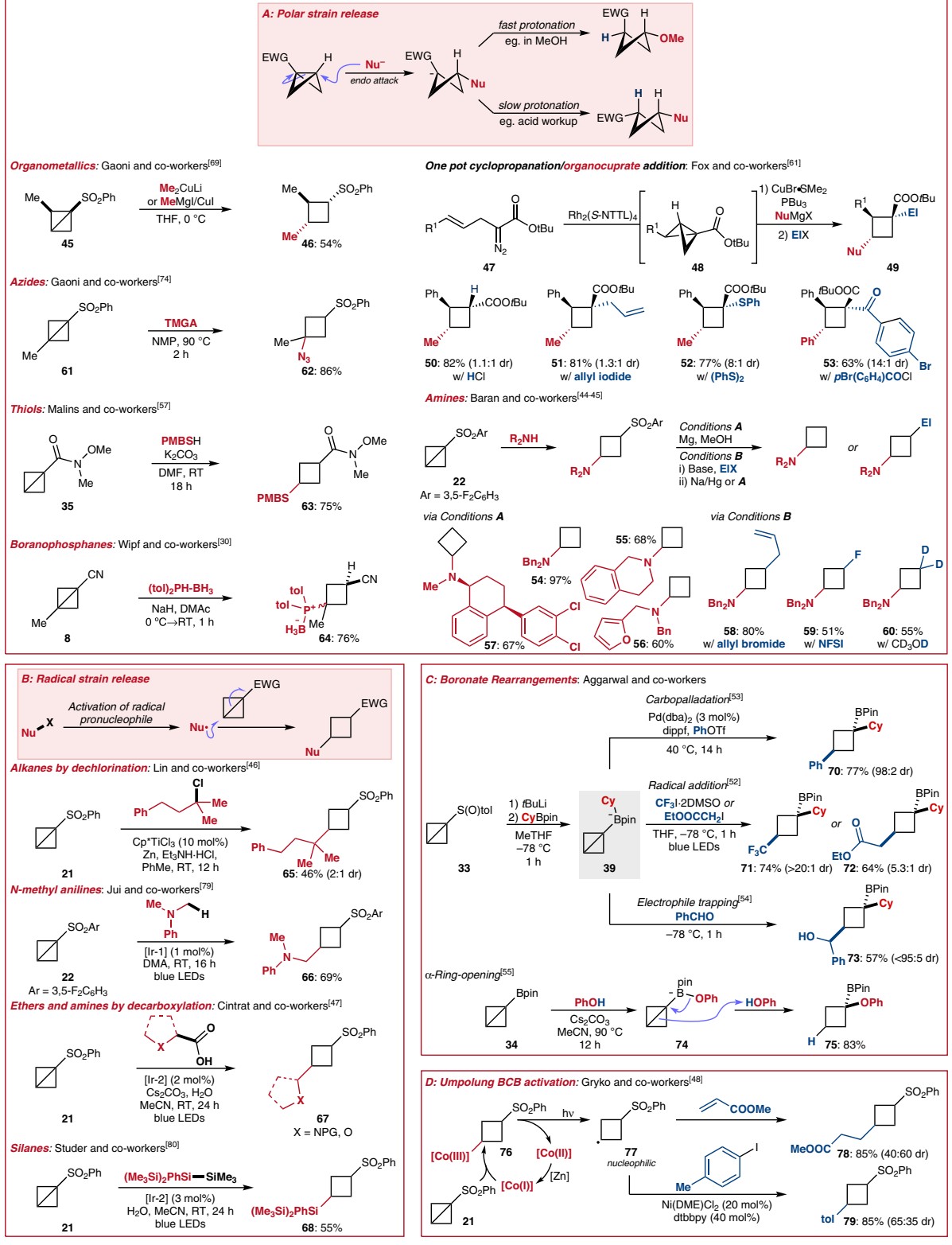

**Fig. 3 Strain-release-driven ring-opening of BCBs to give substituted cyclobutanes. A** Polar strain-release reactions with polar nucleophiles. **B** Radical strain-release reactions with radical nucleophiles. **C** Strain-release reactions driven by boronate rearrangement chemistry. Both polar and radical electrophiles can be used in these reactions. **D** Umpolung BCB activation with Co(I) complexes. Formal nucleophilic attack of the Co(I)-complex into the bicyclobutane gives a Co(III)-intermediate from which a cyclobutyl radical may be photochemically generated. Nu nucleophile, El electrophile, EWG electron-withdrawing group, TMGA tetramethylguanidinium azide, PMB *para*-methoxy benzyl, tol tolyl, NTTL *N*-1,8-naphthaloyl-(*S*)-*tert*-leucine, NFSI *N*-fluorobenzenesulfonimide, [Ir-1] [Ir{dF(CF₃)ppy}₂(dtbbpy)]PF₆, [Ir-2] [Ir(dF(CF₃)ppy)₂(5,5'-d(CF₃)bpy)]PF₆, dba dibenzylideneacetone, dippf 1,1'-bis(diisopropylphosphino)ferrocene, Tf trifluoromethylsulfonyl, pin pinacol, DME dimethoxyethane, dtbbpy 4,4'-di-tert-butyl-2,2'-dipyridyl; Cp* 1,2,3,4,5-pentamethylcyclopentadienyl.

BCB unit is inverted and it is made nucleophilic, reacting with typical electrophiles or electrophilic radicals. Additionally, nucleophile and electrophile addition are simultaneous, eliminating the possibility of diastereomer formation. Boronate 69 is easily accessed from sulfoxide 33 by Li–S exchange and addition of the formed Li-BCB to a boronate ester. Boronate 69 has been shown to engage in Pd-catalysed cross-coupling reactions with organotriflates to yield boranyl cyclobutanes such as 70[53]. BCB 67 acts as the nucleophile in this cross-coupling, with palladium pre-complexation with the central BCB C–C bond triggering boronate rearrangement. Conventional electrophiles may trigger similar boronate rearrangement and be concurrently trapped by BCBs. For example, benzaldehyde was reacted with BCB 67 to afford alcohol-substituted cyclobutane 73[54]. As with other BCBs, boronate 67 may also accept radicals. Electrophilic trifluoromethyl and α-carbonyl radicals could be generated by photolysis of C–I bonds and led to cyclobutanes such as 71 and 72[52]. While most of the shown strain-release ring-opening reactions resulted in the formation of 1,3 difunctionalised cyclobutanes, the treatment of boronic ester-substituted BCB 34 with alcohols, thiols or sulfonamides in the presence of a base resulted in α-functionalisation to give the corresponding *gem*-disubstituted boranocyclobutanes[55]. For example, addition of phenol to boronic ester 34 gave boronate 74 and subsequent protonation of the central C–C bond triggered rearrangement to cyclobutane 75.

An alternative mode of BCB activation was disclosed by Gryko and co-workers, who used a vitamin $B_{12}$-derived Co-corrin to generate nucleophilic cyclobutyl radical 77 (Fig. 3D)[48]. In this approach, the Co(I) catalyst adds to BCB 21 to give the Co(III)-complex 76. Visible light irradiation leads to homolytic Co–C cleavage and formation of radical 77. This can react in both Giese type addition reactions to alkyl-substituted cyclobutanes such as 78 or in reductive Nickel catalysed cross-coupling reactions to give aryl-substituted cyclobutanes such as 79.

### Bridged bicyclic compounds by insertion
Another class of reactions with BCBs are those that involve additions of small fragments across the central BCB C–C bond. These reactions are becoming better understood and more widely appreciated as a unique synthetic entry to bridged bicyclic architectures. The insertion of carbenes to BCBs gives bicyclo[1.1.1]pentane (BCP)-type compounds that are usually accessed via nucleophile addition to [1.1.1]propellane (Fig. 4)[10]. While the latter method allows for versatile bridgehead modifications, the functionalisation of the methylene bridges, for example with halogens, remains challenging. The halogenated BCP products are sought after due to their potential use as bioisosters for substituted benzene rings[81]. The insertion of halogenated carbenes into BCBs provides one route for their synthesis.

The first synthetically useful example of this idea was reported by Applequist and co-workers, who were able to insert dichlorocarbene into BCB 80 to afford BCP 81[82,83]. Dichlorocarbene could be generated from sodium trichloroacetate by thermal decomposition and was also able to react with BCB 15 to give aryl-substituted BCPs. Recently, Mykhailiuk and co-workers reported the insertion of bromofluorocarbene to afford BCP 83[84]. Through C–Br reduction with Raney-Ni, they were also able to obtain the monofluorinated BCP 84. The best investigated version of this reaction is the insertion of difluorocarbene to give the corresponding difluorinated BCPs. Three different sources of difluorocarbene have been reported to be effective. Ma and co-workers first used trimethylsilyl 2-fluorosulfonyl-2,2-difluoroacetate (TFDA)[36,38], with Mykhailiuk and co-workers later using the Ruppert-Prakash reagent, $TMSCF_3$[37]. Recently, Anderson and co-workers reported a few examples with

(triphenylphosphino)difluoroacetate (PDFA)[33,34]. So far, this method is limited to BCBs bearing electron-withdrawing groups (esters or amides) and an aryl (to 85) or vinyl substituent (to 86). Downstream modification of the products can deliver a greater variety of compounds including carboxylic acid 87 and amide 88. The mechanism of the carbene insertion was for a long time not well understood. Early calculations suggested that attack of the C–C σ* orbital by the carbene was the favoured reaction between carbenes and BCBs, but led only to diene-type products[85]. A concerted (cycloaddition-type) mechanism involving attack from above the C–C bond that would lead to the BCP was found to be less favoured. Recently, Anderson and co-workers reinvestigated the mechanism using deuterium-labelled BCB 89[34]. They proposed an alternative, stepwise mechanism based on their experimental observations. The difluoro (singlet) carbene approaches from the bottom face of the BCB to generate zwitterionic intermediate 90 after electrophilic addition. Ring-flip to 91 then allows for cyclisation to anti-substituted BCP 92 (no syn-substituted 92 was isolated). 1,4-Dienes 93 and 94 were also observed as fragmentation side products originating from 90, with the geometry of alkene 94 (highlighted in red) being consistent with bottom and not top-face attack of the carbene. However, the authors did not completely rule out a concerted mechanism, through which the observed fragmentations could also be explained.

A typical reaction of alkenes is the cycloaddition reaction. The proposed π-type character of the central C–C bond of BCBs has led to suggestions that it could also take part in cycloaddition-type chemistry. Importantly, an alkene insertion or [2σ-2π]-type reaction would lead to the formation of bridged bicyclic bicyclo[2.1.1]hexane (BCH) derivatives, another class of proposed benzene bioisosteres (Fig. 5A). Cairncross and Blanchard first investigated this idea, and disclosed the reaction between BCB 8 and butadiene under thermal conditions to give BCH 95[86]. de Meijere and co-workers later reported a similar reaction with captodative alkenes to give bridged compounds such as BCH 96[87]. BCH 97 was also isolated as a side-product in some reactions; they reasoned that its formation involved the rearrangement of BCB 8 to the corresponding butadiene and subsequent cycloaddition of this butadiene with remaining BCB 8. Wipf and Walczak also reported an intramolecular version of this type of reaction to give access to *aza*-polycyclic architectures such as 98[65]. Generally, these thermal reactions require high temperatures and lead to only low isolated yields, mostly due to thermal instability of the BCBs.

Leitch and co-workers went beyond the use of alkenes and reported the insertion of imines into BCBs in a Lewis acid-catalysed stepwise cyclisation to *aza*-BCHs (Fig. 5B)[88]. BCBs bearing both a carbonyl and aryl group were necessary for the reaction, but a range of typical aromatic and heteroaromatic systems were tolerated (as in 99 and 100) as were amide substituents (as in 101). The reaction was proposed to proceed in a stepwise fashion; after coordination of Gd(OTf)₃ to the carbonyl group of the BCB, enolate formation is induced. Subsequent attack into the imine occurs, and cyclisation of the formed amide onto the intermediate carbocation affords the *aza*-BCH scaffold. At room temperature and with an *N*-aryl group, the basicity of the amide was tempered sufficiently for cyclisation to *aza*-BCH 99 to be the dominant reaction. In the case of the *N*-alkyl amide, cyclobutene 102 was obtained upon heating due to the higher basicity of the amide and dominance of the deprotonation side-reaction.

Malins and co-workers investigated the cycloaddition of triazolinediones (TADs, 103) to BCBs to yield BCB-TAD adducts including 104–106 (Fig. 5C)[89]. The reaction proceeds under ambient reaction conditions, requires no additives, and is fully

**Fig. 4 Carbene insertions into BCBs to give halogenated BCPs. A** Insertion of dichlorocarbene into bicyclobutanes. **B** Insertion of bromofluorocarbenes radicals into bicyclobutanes and the selective debromination to give monofluorinated bicyclopentanes. **C** Insertion of difluorocarbenes into bicyclobutanes along with functional group tolerance and downstream targets from these products. **D** Mechanistic investigations into the carbene insertion reaction by Anderson and co-workers which support a stepwise insertion mechanism. EDA ethylenediamine, TMS trimethylsilyl, Boc *tert*-butoxycarbonyl.

atom-economic. The reaction tolerates a variety of electronically distinct TADs (**104–105**). By linking biomolecules to the TAD unit (as in **106**), the authors also demonstrated the potential of this method for bioconjugation.

The insertion of alkenes into BCBs has recently been reinvestigated, now making use of advances in photocatalysis to overcome the limitations caused by the thermal instability of BCBs (Fig. 5D). Two complementary approaches towards BCHs have been disclosed by Glorius and Brown. Glorius and co-workers made use of thioxanthone as a photocatalyst to sensitise the alkene reaction partner. The excited state alkene then reacted with a range of mostly monosubstituted BCBs through a formal [2π + 2σ] cycloaddition to deliver a range of BCHs[90]. Coumarins (to **107**, **109**, and **110**), indoles (to **108**), and even imine derivatives (to *aza*-BCH **111**) participated. Disubstituted BCBs reacted with opposite regioselectivity (compare **107** with **109**) and a boronic ester-substituted BCB was also tolerated to give BCH **110**. The latter is potentially interesting as a starting material for cross-coupling reactions. Instead of sensitising the alkene, Brown and co-workers devised a method to sensitise the BCB itself, prior to alkene insertion[91]. This was accomplished using naphthylketone-substituted BCBs (e.g. **116**). The naphthyl group was key to enabling sensitisation by a thioxanthone-based photocatalyst to give **116**\*. This excitation triggered the homolytic cleavage of the central C–C bond to afford diradical **117**. A suitable alkene was attacked by the less-stabilised radical position and radical recombination completed the stepwise cyclisation to BCH **119**. Monosubstituted BCBs (to **119**) as well as disubstituted BCBs (to **112**) were shown to react with styrene. In addition, 1,1-disubstituted alkenes (to **113**), and alkenes bearing boronic esters (to **114**) and heteroaromatic substituents (to **115**) were also tolerated.

A related method was reported by Procter and co-workers (Fig. 5E)[92]. They devised a SmI₂-catalysed redox reaction to control the insertion of the alkene reaction partners into the BCB. Sm(II)I₂ is a potent single-electron-reductant and readily reduced the carbonyl group of ketone **125** to afford first the corresponding ketyl radical, and then via homolytic cleavage of the C–C bond, cyclobutyl radical **126**. Giese addition to electron-deficient alkenes followed by radical rebound and back electron transfer to Sm(III) delivered BCH **120**. Mostly alkyl ketones are used, but

aryl ketones (to **121**) also react. Disubstituted BCBs (to **122**) are also tolerated. A range of terminal alkenes were inserted, including acrylonitrile (to **120–121**), acrylates (to **123**), and vinyl sulfones (to **124**), to deliver a library of BCHs bearing a variety of useful functional handles.

## Fused and spirocyclic compounds by rearrangement

A comparatively less investigated, but potentially powerful use of BCBs is in rearrangement chemistry. Wipf and co-workers have conducted investigations in this area, and developed routes to both fused and spirocyclic scaffolds under a wide range of reaction conditions (Fig. 6). Particularly useful is the divergent synthesis of bicyclo[3.1.0]hexane or bicyclo[5.1.0]octane fused scaffolds by Rh(I)-catalysed cycloisomerisation, with the selectivity governed by the phosphine ligand used (Fig. 6A)[58]. Beginning from allyl amine-substituted BCB **128**, bicyclo[3.1.0] octane **129** was obtained when using PPh₃, and bicyclo[5.1.0] octane **130** was formed when using dppe. A small library of compounds was prepared, including derivatives with aliphatic substituents (as in **131** and **132**), ethers (as in **133** and **134**), and heterocycles (as in **135** and **136**). One allyl ether was also subjected to the reaction conditions. In this case, bicyclo[5.1.0]octane **138** was successfully obtained but THF **137** was formed instead of the expected bicyclo[3.1.0]octane; ring-opening of the cyclopropane occurred to give the diene motif. Mechanistically, the reactions to the two distinct products were proposed to follow similar pathways. Initial insertion of Rhodium into the central C–C bond of the BCB led to rhodabicycle **139** and a formal retro-[2 + 2] cycloaddition led to one of two metal carbenes, **140** or **141**, depending on the ligand used. Intramolecular cyclopropanation provided the fused [n.1.0] products **129** and **130**. Glorius and co-workers reported another application of rhodium-catalysed rearrangements of BCBs, cleaving both an internal and external C–C bond of the BCB to give linear alkene derivatives (not shown)[56]. Wipf and co-workers found that allyl amine-substituted BCBs similar to those used previously also underwent thermal Alder-Ene reactions to give [4.3] spirocyclic compounds such as **144** (Fig. 6B)[93]. Beginning from BCB **142**, allylation under basic conditions led to allyl amine **143** which underwent an in situ Alder-Ene reaction at slightly elevated temperature via a

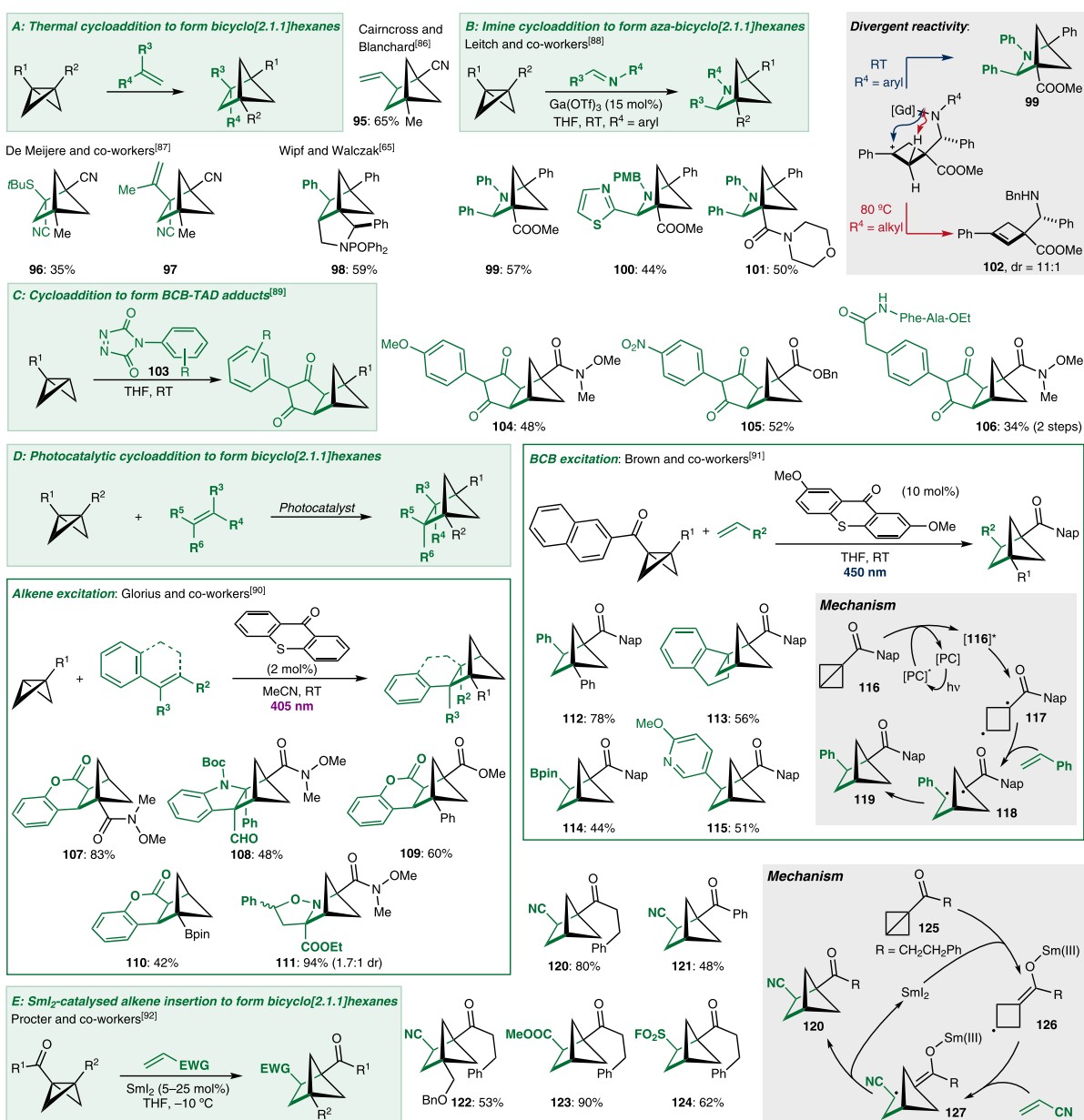

**Fig. 5 Alkene and imine insertions into BCBs to give bridged bicyclic bicyclo[2.1.1]hexanes. A** Thermal cycloaddition reactions between bicyclobutanes and alkenes to form bridged bicyclic structures. **B** Lewis acid-catalysed cycloaddition reactions between bicyclobutanes and imines to form *aza*-bridged bicyclic structures. **C** Cycloaddition of bicyclobutanes with triazolinediones and applications in peptide modification. **D** Cycloaddition of bicyclobutanes with alkenes via energy transfer photocatalysis to form bridged bicyclic structures. **E** SmI₂ catalysed insertion of alkenes into bicyclobutanes and the proposed mechanism. Tf trifluoromethylsulfonyl, PMB *para*-methoxy benzyl, TAD triazolinedione, Phe phenylalanine, Ala alanine, Boc *tert*-butoxycarbonyl, pin pinacol, Nap naphthyl, [PC] photocatalyst, EWG electron-withdrawing group.

transition state that led to anti-substitution on spirocyclic pyrrolidine **144**. This reactivity was demonstrated for two other substrates. Replacing the acrylate moiety of **143** with a pendant propargyl alcohol led to spirocycle **145**, and an α,β-unsaturated aldehyde provided spirocycle **146**, which was used as an intermediate en-route to the tricyclic core of daphniglaucin natural products. Recently, Wipf and co-workers disclosed an alternative approach to spirocycles from BCBs, preparing a range of [4.3] spirocyclic compounds[94]. Exploiting the acidity of the bridgehead C–H bond, they were able to deprotonate BCB **21** with *n*BuLi, and add the formed organolithium species into cyclobutanone to give alcohol **147**. Addition of acid led to a suprafacial Semipinacol rearrangement of **147** to spirocycle **148**. The relative stereochemistry was confirmed by X-ray crystallography. Heterocyclic

products such as **149** are accessible and, as expected, the more substituted carbon atom rearranges, leading to congested adjacent stereocentres as in **150** and **151**. Addition into 5, 6, and 8-membered cyclic ketones with subsequent rearrangement was also possible, as was the use of *N*-bromosuccinimide in place of Brønsted acid to initiate the Semipinacol rearrangement. In this way, brominated [5.3] spirocycle **152** was obtained.

## Highly substituted BCBs by directed metallation
Functionalisation of the BCB scaffold provides the opportunity to expand the utility of the reactions described above by providing more densely substituted BCBs as starting materials. Although a rather neglected area of BCB research, interest in this topic has

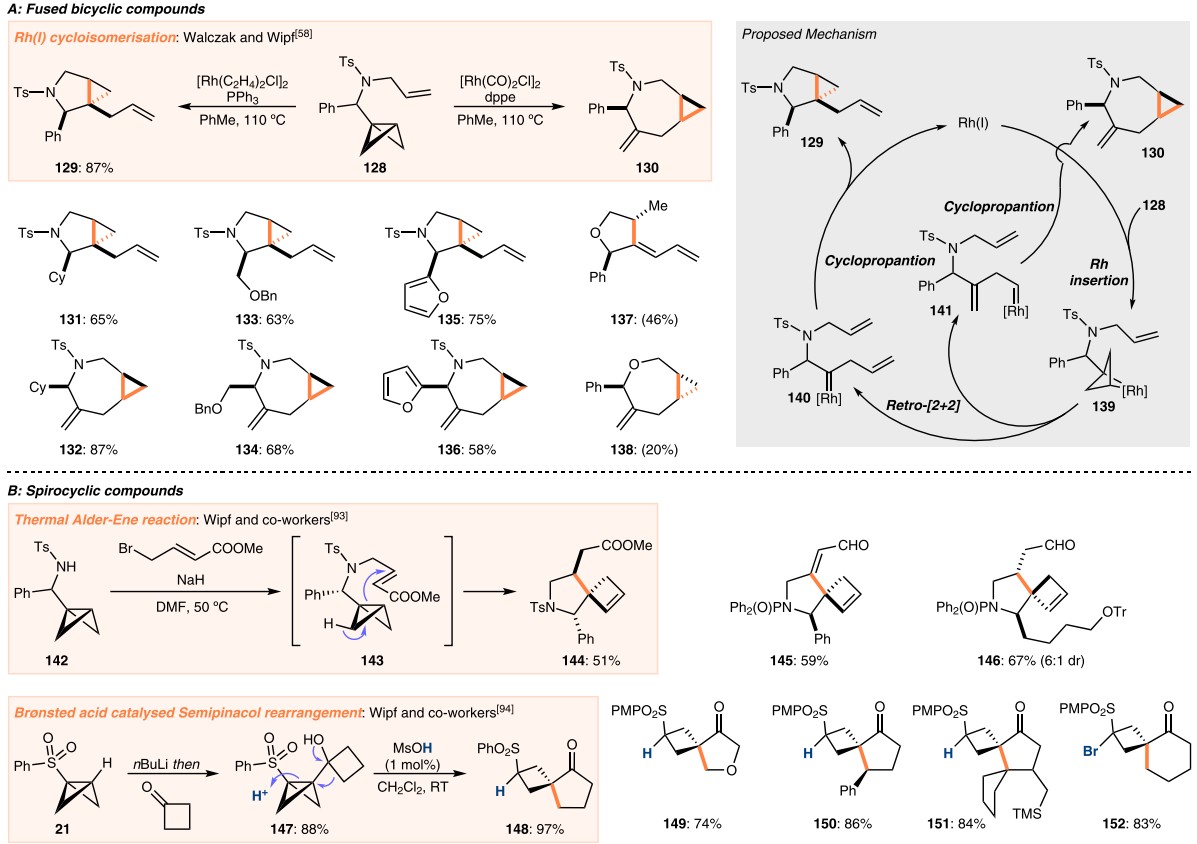

**Fig. 6 Rearrangement reactions of BCBs to give fused and spirocyclic compounds. A** Divergent Rh(I)-catalysed cycloisomerisation reactions of bicyclobutanes to give fused bicyclic structures. The selectivity of the reaction can be controlled by the choice of ligand and the initial step of the reaction involves the insertion of the rhodium into the bicyclobutane. **B** Reactions of bicyclobutanes that lead to spirocyclic compounds. Both a thermal Alder-Ene reaction and an acid-catalysed Semipinacol rearrangement are highlighted. Yields in brackets refer to yields determined by [1]H NMR. dppe 1,2-bis(diphenylphosphino)ethane, Ts toluene sulfonyl, Ms methane sulfonyl, PMP *para*-methoxyphenyl.

recently been revived. In early examples of this reactivity, Gaoni and co-workers harnessed the acidity of the bridgehead C–H bond to prepare bridgehead-substituted BCB scaffolds (Fig. 7A)[95,96]. Starting from aryl sulfone-substituted BCBs, deprotonation of the bridgehead C–H bond with *n*BuLi led to Li-BCB **153** which could in turn be trapped by a range of typical electrophiles. Aldehydes (to **154**), alkyl halides (to **155**), chloroformates (to **156**), and epoxides (to **157**) could all be reacted with Li-BCB **153**. Recently, Anderson and co-workers exploited this deprotonation to develop Negishi cross-coupling reaction conditions which now provide access to aryl- and vinyl-substituted BCBs[33]. Lithiation with PhLi and transmetallation with ZnCl₂ provided Zn-BCB **158**, which could be used directly in palladium-catalysed cross-coupling reactions with vinyl and (hetero)aryl iodides. Electron-rich and electron-poor aryl iodides could be used, for example 2-iodobenzotrifluoride to BCB **159**. Vinyl iodides reacted with retention of geometry to give BCBs such as **160**. Additionally, heteroaryl iodides including pyridines, quinolines (to **161**), isoquinolines, and indoles were all tolerated. Sulfonamide-substituted BCBs could also be used instead of the aryl sulfone (not shown). Directed bridge functionalisation of BCBs is much harder to achieve. Anderson and co-workers reported one successful strategy which involved blocking the bridgehead position with a substituent (Fig. 7B)[34]. This time using an amide-substituted BCB with a methyl group at the bridgehead, lithiation occurred at a bridge position to give Li-BCB **162**. To offset the reduced acidity of the bridge C–H bonds, *s*BuLi/TMEDA was needed in place of *n*BuLi. Typical electrophiles such as alkyl halides (to **163** and **167**) and chloroformates

(to **164**) could react with Li-BCB **162**. In addition, a range of heteroatom electrophiles were used. Boronic acid **165** could be accessed by trapping **162** with trimethyl borate and silanes, thioethers, phosphines, stannanes, and germanes were also all accessible (not shown). Sequential functionalisation of both bridge positions was also possible; starting from BCB **167**, ether **166** was obtained by reaction with MOMCl. Alternative bridge-head blocking groups can be also used, including silanes (as in **168**) and aryl groups (as in **169**). Finally, the authors were able to achieve bridge desymmetrisation through a diastereoselective silylation of BCB **170**. The chiral amide substituent directed diastereoselective lithiation trapping with chlorotrimethylsilane gave silane **171** in 2.7:1 dr.

## Outlook
With the advent of improved and tractable synthetic routes to BCBs, investigations into their unique reactivity have led to BCBs becoming powerful building blocks for the synthesis of functionalised cyclobutanes and bridged bicyclic compounds. Initial investigations show that they also offer an unusual entry to fused and spirocyclic compounds, but this remains relatively untapped and should be further investigated. The insertion of carbenes to access bridge-substituted BCP derivatives is potentially a powerful tool towards some unusual benzene bioisosteres but at the moment this chemistry is limited to halocarbenes; expansion of this method to other carbene derivatives would be extremely useful. Likewise, transition-metal reactivity with BCB units remains underdeveloped and might offer access to reaction pathways orthogonal to those that already exist. Recent

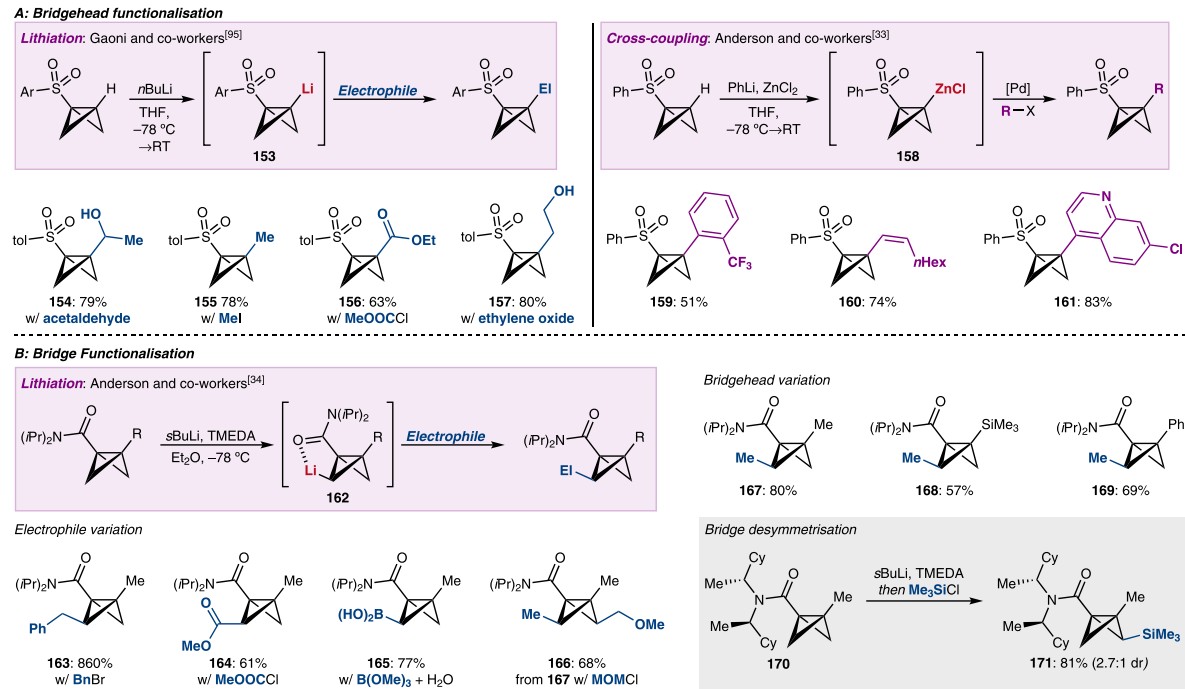

**Fig. 7 Functionalisation reactions of BCBs to give highly substituted BCBs. A** Functionalisation of the bridgehead position of bicyclobutanes by deprotonation with strong organolithium bases. The intermediates can then be trapped with typical electrophiles or, after transmetallation to zinc, be used in Pd-catalysed cross-coupling reactions. **B** Functionalisation of the bridge positions of bicyclobutanes by deprotonation with strong bases. Desymmetrisation of the bridge positions is also possible with chiral amide starting materials, leading to chiral bicyclobutane structures. El electrophile, Ar aryl, tol tolyl, TMEDA tetramethylethylenediamine, MOM methoxymethyl.

developments in the synthesis of more highly functionalised BCBs should also enable more detailed investigations of these scaffolds and the effect that alternative substitution has on BCB reactivity. This is a rapidly developing field and during the review process of this manuscript further investigations towards bridged bicyclic[97,98], spirocyclic[99], and other structural motifs[100,101] have been reported. As the preparation of structurally complex architectures becomes more important to our ability to design the next generation of pharmaceuticals, the chemistry of the smallest fused hydrocarbon will certainly have its part to play. We look forward to seeing further exciting developments in the future!

## Data availability

No datasets were generated or analysed during the current study.

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

## Acknowledgements

We wish to acknowledge the Georg-August-Universität Göttingen for financial support. J.C.L.W. wishes to thank Prof. Manuel Alcarazo for his continuous generous support and guidance.

## Author contributions

J.C.L.W. conceived the scope of the article. Both authors discussed the content, and wrote and edited the manuscript.

## Competing interests

The authors declare no competing interests.
