## [Peer Review File · Communications Chemistry]

Reviewers' comments:

Reviewer #1 (Remarks to the Author):

Summary:

The authors provide an overview of the different options present in the literature for the synthesis of bicyclobutane (BCB) moieties, the different classes of reactivity of bicyclobutanes, as well as explore the main classes of compounds that can be synthesized from bicyclobutane and its derivatives. The authors provide a complete review of the initial synthetic investigation, structure, and reactivity of BCB. This foundation sets the stage for the review of current synthetic approaches to BCB. It is recommended the authors include the first reference listed below. The review continues with reactions of BCB to form substituted bicyclobutanes. Within the section titled: Bridged bicyclic compounds by insertion, it is recommended the authors include the second reference listed below. Overall, the authors have established a reasonably comprehensive review of BCB and the applicable chemical space surrounding BCB moieties.

References to include:

Tokunga, K., Sato, M., Kuwata, K., Miura, C., Fuchida, H., Matsunaga, N., Koyanagi, S., Ohdo, S., Shindo, N., Ojida, A. Bicyclobutane Carboxylic Amide as a Cysteine-Directed Strained Electrophile for Selective Targeting of Proteins. *J. Am. Chem. Soc.* 142. 18522–18531 (2020).

Schwartz, B. D., Smyth, A. P., Nashar, P. E., Gardiner, M. G., Malins, L. R. Investigating Bicyclobutane– Triazolinedione Cycloadditions as a Tool for Peptide Modification. *Org. Lett.* 24, 1268–1273 (2022).

Some specific edits and other recommendations:

Page 3, Figure 1: It will improve clarity to show the actual compound synthesized in the route from compound 2 to 3.

Page 3, Figure 1, Insertion reactions; The R group are both not labeled and labeled as R1 and R2, in the same figure.

Page 3, Paragraph 1: The language "...it was too long and laborious to be of real practical use" is overly casual.

Page 5, Figure 2, Path B: One-pot synthesis...; "Jung and Lindsay" has superscript XX which should read 49

Page 5, Figure 2, Path C: Side-chain...; Compound 40 has a text justification issue with the XH substituent where the carbon bond is attached at the H not X.

Page 7, "H-phosphonates" should be written as "H-Phosphonates" with H italicized and P capitalized

Page 8, Figure 3, Ethers and amines...; PG is undefined on X=NPG... it may be helpful to define in either Figure or text.

Reviewer #2 (Remarks to the Author):

This Communications Chemistry review article by Walker and co-workers covers the physical features,

preparation and reaction of the highly strained bicyclo[1.1.0]butane (BCB) unit from its initial discovery to recent work. Although this review is one of the latest in a series of recent reviews on the same topic, such as references 9-11, its inclusion of the most recent cycloaddition chemistry, and in general how comprehensive it is in covering the BCB field, makes it a very valuable one.

The manuscript is of high quality and contains an excellent summary of all the BCB literature. Below is listed a few minor comments I think the authors should address in their manuscript prior to publication:

- The authors state that 'the bridgehead C-H bond ... is ... more acidic than a conventional C-H bond and can be deprotonated with strong organometallic bases'. Is the pKa of this C-H bond known?
- I would make it clearer within Scheme 1B that the reactions shown do not produce the products shown; I do not think that simply putting 'proposed structure' sufficiently conveys the fact that these products are incorrect. In the case of the example by Perkin and Simonsen, I would also include the actual product formed.
- The work by Aggarwal is noted as 'distinct' in the seventh paragraph of the 'Substituted cyclobutanes by strain-release ring opening' section, but this distinction is not explicitly explained. The difference is that the BCB unit is rendered nucleophilic (reacting with many standard 2-electron electrophiles as well as electrophilic radicals) whereas in all other chemistries the BCB unit is electrophilic (reacting with many standard 2-electron nucleophiles as well as nucleophilic radicals). Also, as the addition of the nucleophile (the Cy group as drawn) is simultaneous with reaction with the electrophile, there is no chance of formation of minor diastereomers. This arguably makes it not strictly fit into the generic reaction profile at the top of Figure 3.
- The other seeming outlier is the work from Gryko, which the authors highlight as being distinct from other works as it '[generates a] nucleophilic cyclobutyl radical'. While this is true, the chemistry of the cyclobutyl radical is not the chemistry of the BCB unit. An inspection of Gryko's work reveals that (and I accept that the mechanistic work is not extensive) the generation of the cyclobutyl radical from the BCB precursor actually relies on the expected reactivity of electrophilic BCB, from reference 48: 'in the presence of a proton source the nucleophilic Co(I) form of the catalyst would undergo conjugate addition to the pi-like central C-C bond of BCB, furnishing Co(III)-alkyl complex'. Therefore, although this work is interesting, I would argue it is interesting in how the Co catalyst interacts with the BCB unit, not because the cyclobutyl radical it generates is nucleophilic.
- The titles for the various sections of Figure 5 read as, for example, 'Thermal cycloaddition to bicyclo[2.1.1]hexanes'. This could read as the BCB unit, which is the topic of the review, is undergoing a thermal cycloaddition reaction with a bicyclo[2.1.1]hexane unit; this is of course not the case. I would recommend something like 'thermal cycloaddition to form bicyclo[2.1.1]hexanes' instead.
- In paragraph 4 of the 'Bridged bicyclic compounds by insertion' section, it is stated that the Leitch chemistry 'proceed[s] in a stepwise fashion', but this mechanism is not elaborated on in the text; I recommend a short sentence to explain this in some more detail and link it to the depiction in Figure 5.
- In paragraph 5 of the 'Bridged bicyclic compounds by insertion' section, the thioxanthone photocatalysts are given abbreviations which are never later referred to.
- It strikes me that work by Glorius cited as reference 56 is not given any discussion.
- I would personally like to see citations within the schemes so that it is easier to, for example, skim each scheme for a product you like and then find the relevant publications without having to go through the text. Likewise, no yields are included despite specific reaction conditions/products being given; there are some instances where low yields are mentioned, but no actual yield is given in either the text or scheme. I would personally like to see yields within the schemes.
- The 'outlook' section is more like a summary than an inspiring outlook, in my opinion. I would suggest adding some more imaginative suggestions for future work, such as (and I am not saying these should be included, merely that they could) further investigating the reaction of BCBs with transition metals and radicals, a greater range of carbenes/carbenoids, and checking whether there

are any quirks in the chemistry of 1,3-disubstituted/bridge functionalised BCBs.

Overall, I think this is a valuable and comprehensive article on BCBs that warrants publication in Communications Chemistry following minor corrections. I hope you find these comments useful, and good luck with your (and I am presuming here) future research into this field.

Reviewer #3 (Remarks to the Author):

Recently, BCB and its derivatives have been in the centre of interest of organic chemistry, particularly those functionalized at the bridgehead carbon atoms, and the field has been recently a subject of many excellent reviews, including one from 2022 (Chem Sci, 2022, 13, 11721-11737) Strained molecules are valuable building blocks for the synthesis of functionalized bicyclo[1.1.1]pentanes, cyclobutanes, high value bioisosteres. These molecules, up to date, as electrophiles, were shown to undergo nucleophilic (P. Baran et al. Science 2016, 351, 241) and radical (J. Nugent et al. ACS Catal. 2019, 9, 9568) addition and can be functionalized via cyclopalladation reaction (S. Clementson, et al. Org. Lett. 2019, 21, 4763). So there is a question of whether there is room for a new and up-to-date one. Because the field is evolving rapidly, the review by Walker and co-workers providing an overview on the synthetic applications of BCB to achieve structurally complex scaffolds may be of interest to organic and medicinal chemists. It is a useful piece of information on the structure, synthesis, and reactivity of bicyclobutanes.

The manuscript needs to be reviewed to eliminate some inaccuracies to help the reader to better follow the discussion of the described data. I suggest the following:

1. I do not fully see the rationale behind the organization of the chapters in this manuscript. The paper starts with a critical discussion of the structure of BCB that is followed by 'initial synthetic investigations'. In my opinion, this is just a description of the very first unsuccessful attempts to synthesize the bicyclic scaffold. Although I understand why the authors wanted to mention them, I do not see the point of giving schemes showing reactions with false products. Figure 1B should be removed from the manuscript, it is quite misleading. In the opinion of this reviewer, the paragraph should be included in the section 'Synthesis of bicyclobutanes'. In the next part, the general reactivity of BCBs is presented. Why, it is not preceding paragraphs on reporting synthesis of complex structures from BCBs.
2. While writing about the BCB reactivity, (page 2), the authors wrote 'This has been broadly exploited, including nucleophilic and electrophilic addition, and insertion-type reactions.' The reactivity of BCBs in radical reactions should be mentioned separately, even though radical intermediates can have nucleophilic and electrophilic character.
3. I do not agree that Reviews 9,10,11 focus mainly on nucleophile addition reactivity. The description is not correct.
4. Include yields on schemes to help readers understand how efficient the method is.
5. The authors wrote: 'the naphthyl group was sensitized....' I would rather argue that the compound was sensitized and not a group.
6. The authors wrote 'a vitamin B12-derived Co-porphyrin' This statement is incorrect. Vitamin B12 is based on a corrin ring and not a porphyrin ring.
7. Similarly, 'Sm(II) is a potent single-electron-reductant' I would rather put here Sm(II)I₂ as a reductant.
8. Figure 5 – box showing Brown's work. In the mechanism, olefin is missing.
9. The manuscript is written in a too sophisticated, fancy English (unnecessary), just to name a few: a rich tapestry of complex molecules; BCB structure began to reveal itself; have begun to bear fruit.

Author Responses to Reviewers:

Reviewer 1 (Changes to manuscript highlighted in yellow)

- It is recommended the authors include the first reference listed below. " Tokunga, K., Sato, M., Kuwata, K., Miura, C., Fuchida, H., Matsunaga, N., Koyanagi, S., Ohdo, S., Shindo, N., Ojida, A. Bicyclobutane Carboxylic Amide as a Cysteine-Directed Strained Electrophile for Selective Targeting of Proteins. *J. Am. Chem. Soc.* 142. 18522–18531 (2020)." We thank the reviewer for highlighting this reference which we had originally missed. The indicated reference has been added (Ref. 76) along with a short discussion.
- The review continues with reactions of BCB to form substituted bicyclobutanes. Within the section titled: Bridged bicyclic compounds by insertion, it is recommended the authors include the second reference listed below. "Schwartz, B. D., Smyth, A. P., Nashar, P. E., Gardiner, M. G., Malins, L. R. Investigating Bicyclobutane– Triazolinedione Cycloadditions as a Tool for Peptide Modification. *Org. Lett.* 24, 1268–1273 (2022)." We thank the reviewer for highlighting this reference which we had originally missed. The indicated reference has been added (Ref. 89) along with a discussion. The described chemistry has also been added to the figure.
- Page 3, Figure 1: It will improve clarity to show the actual compound synthesized in the route from compound 2 to 3. As all reviewers commented on this, we have elected to remove the relevant section of Figure 1 entirely and move the now shorter discussion to the beginning of the section titled "Synthesis of Jun.-Prof. Dr. Johannes Walker Tel. +49 (0)551 / 39-23768 johannes.walker@chemie.uni-goettingen.de Dr Huijuan Guo Associate Editor, Communications Chemistry Heidelberger Platz 3 14197 Berlin 2 bicyclobutanes". All references remain. The reaction scheme for the first successful synthesis of a bicyclobutane by Wiberg and Ciula has been incorporated into Figure 2.
- Page 3, Figure 1, Insertion reactions; The R group are both not labeled and labeled as R1 and R2, in the same figure. Groups are now labelled R1 and R2 consistently where appropriate.
- Page 3, Paragraph 1: The language "...it was too long and laborious to be of real practical use" is overly casual. We have removed this sentence and modified the subsequent sentence slightly to "Much work since the pioneering synthesis of BCB 5 by Wiberg and Ciula has been devoted to establishing more efficient methods for BCB synthesis."
- Page 5, Figure 2, Path B: One-pot synthesis...; "Jung and Lindsay" has superscript XX which should read 49 Thank you for pointing this out. The reference has been added.
- Page 5, Figure 2, Path C: Side-chain...; Compound 40 has a text justification issue with the XH substituent where the carbon bond is attached at the H not X. Thank you for pointing this out. The text justification has been corrected.
- Page 7, "H-phosphonates" should be written as "H-Phosphonates" with H italicized and P capitalized Thank you for pointing this out. The typographical changes have been made.

- Page 8, Figure 3, Ethers and amines...; PG is undefined on X=NPG... it may be helpful to define in either Figure or text. Thank you for this suggestion. We have made appropriate additions to the text and agree this is a useful addition.

Reviewer 2 (Changes to manuscript highlighted in green)

- The authors state that ‘the bridgehead C-H bond ... is ... more acidic than a conventional C–H bond and can be deprotonated with strong organometallic bases’. Is the pKa of this C-H bond known? An experimental measurement of the pKa of the C-H bond is, to the best of our knowledge, not published. The closest we could find was in an accessible PhD thesis (Link: <http://dscholarship.pitt.edu/9371/1/MAWalczakPhDThesis.pdf>), which suggested a pKa of 36 based on the 1 JCH coupling constant of 205 Hz. No reference to any source was given and we would feel 3 uncomfortable about giving this value as is in the review. We have therefore omitted all comparison of acidity and rewritten the sentence as "The bridgehead C–H bond also accounts for some of the unusual reactivity of BCBs; it is strongly polarised and can be deprotonated with strong organometallic bases"

- I would make it clearer within Scheme 1B that the reactions shown do not produce the products shown; I do not think that simply putting ‘proposed structure’ sufficiently conveys the fact that these products are incorrect. In the case of the example by Perkin and Simonsen, I would also include the actual product formed. Thank you for this suggestion. Please see our alterations as described above (Reviewer 1).

- The work by Aggarwal is noted as ‘distinct’ in the seventh paragraph of the ‘Substituted cyclobutanes by strain-release ring opening’ section, but this distinction is not explicitly explained. The difference is that the BCB unit is rendered nucleophilic (reacting with many standard 2-electron electrophiles as well as electrophilic radicals) whereas in all other chemistries the BCB unit is electrophilic (reacting with many standard 2-electron nucleophiles as well as nucleophilic radicals). Also, as the addition of the nucleophile (the Cy group as drawn) is simultaneous with reaction with the electrophile, there is no chance of formation of minor diastereomers. This arguably makes it not strictly fit into the generic reaction profile at the top of Figure 3. Thank you for this detailed and helpful comment. We have added some additional discussion in the text on this matter and have also changed the colours of the radical (precursors) in question in the figure to match that of other electrophilic species. Thank you also for the comment regarding the generic reaction profile. This was not intended as a general reaction profile for the whole figure, only for the reactivity described at the top. This comment made it clear to us that this was not obvious! We have now compartmentalised the various reactivities more clearly.

- The other seeming outlier is the work from Gryko, which the authors highlight as being distinct from other works as it ‘[generates a] nucleophilic cyclobutyl radical’. While this is true, the chemistry of the cyclobutyl radical is not the chemistry of the BCB unit. An inspection of Gryko’s work reveals that (and I accept that the mechanistic work is not extensive) the generation of the cyclobutyl radical from the BCB precursor actually relies on the expected reactivity of electrophilic BCB, from reference 48: ‘in the presence of a proton source the nucleophilic Co(I) form of the catalyst would undergo conjugate addition to the pi-like central C-C bond of BCB, furnishing Co(III)- alkyl complex’. Therefore, although this because the cyclobutyl radical it generates is nucleophilic. Thank you for this helpful suggestion. We have amended our discussion appropriately.

- The titles for the various sections of Figure 5 read as, for example, ‘Thermal cycloaddition to bicyclo[2.1.1]hexanes’. This could read as the BCB unit, which is the topic of the review, is undergoing a thermal cycloaddition reaction with a bicyclo[2.1.1]hexane unit; this is of course not the case. I would

recommend something like 'thermal cycloaddition to form bicyclo[2.1.1]hexanes' instead. Thank you for this suggestion. This was also discussed during preparation of the manuscript and we are happy to make this alteration.

- In paragraph 4 of the 'Bridged bicyclic compounds by insertion' section, it is stated that the Leitch chemistry 'proceed[s] in a stepwise fashion', but this mechanism is not elaborated on in the text; I recommend a short sentence to explain this in some more detail and link it to the depiction in Figure 5. Thank you for this suggestion. We agree the original description was not sufficient and have amended the discussion appropriately.
- In paragraph 5 of the 'Bridged bicyclic compounds by insertion' section, the thioxanthone photocatalysts are given abbreviations which are never later referred to. Thank you for pointing this out. We decided that the abbreviations were superfluous and have removed them.
- It strikes me that work by Glorius cited as reference 56 is not given any discussion. Thank you for pointing this out. The chemistry described is a singular outlier and does not provide a method towards a fused, bridged, or spirocyclic compound. Nevertheless, it certainly deserves mentioning and we have added a brief comment in the section of Rhodium-catalysed rearrangements, where it fits best.
- I would personally like to see citations within the schemes so that it is easier to, for example, skim each scheme for a product you like and then find the relevant publications without having to go through the text. Likewise, no yields are included despite specific reaction conditions/products being given; there are some instances where low yields are mentioned, but no actual yield is given in either the text or scheme. I would personally like to see yields within the schemes. We agree that these changes would make the manuscript more useful to the reader and have added references and yields (where they are given by the original authors) to the schemes. Thank you for this suggestion!
- The 'outlook' section is more like a summary than an inspiring outlook, in my opinion. I would suggest adding some more imaginative suggestions for future work, such as (and I am not saying these should be included, merely that they could) further investigating the reaction of BCBs with transition metals and radicals, a greater range of carbenes/carbenoids, and checking whether there are any quirks in the chemistry of 1,3-disubstituted/bridge functionalised BCBs. Thank you for this suggestion. We agree that our original outlook was more of a summary. We have removed some of the more mundane text and added some of the proposed additions.

Review 3 (Changes to manuscript highlighted in blue)

- I do not fully see the rationale behind the organization of the chapters in this manuscript. The paper starts with a critical discussion of the structure of BCB that is followed by 'initial synthetic investigations'. In my opinion, this is just a description of the very first unsuccessful attempts to synthesize the bicyclic scaffold. Although I understand why the authors wanted to mention them, I do not see the point of giving schemes showing reactions with false products. Figure 1B should be removed from the manuscript, it is quite misleading. In the opinion of this reviewer, the paragraph should be included in the section 'Synthesis of bicyclobutanes'. In the next part, the general reactivity of BCBs is presented. Why, it is not preceding paragraphs on reporting synthesis of complex structures from BCBs. The overall structure of the review was agreed with the editorial office beforehand. We wanted to provide a general foundation on BCB properties, reactivity, and synthesis (this was a suggestion of the editorial office)

before concentrating on the opportunities available for complex molecule synthesis. Omitting any of the above topics would have required readers to access additional sources for a complete picture. As discussed above, we decided to remove the scheme with since discredited product characterisations and on the suggestion of the reviewer have moved the discussion and the first successful synthesis of a BCB to the section "Synthesis of bicyclobutanes." We also agree that it is a shame that a general scheme on the types of compounds that are accessible from BCBs was missing from Figure 1. We have now added this along with some accompanying text, and thank the review for this and their other related suggestions.

- While writing about the BCB reactivity, (page 2), the authors wrote 'This has been broadly exploited, including nucleophilic and electrophilic addition, and insertion-type reactions.' The reactivity of BCBs in radical reactions should be mentioned separately, even though radical intermediates can have nucleophilic and electrophilic character. Thank you for this suggestion and agree that we overlooked this in our original manuscript. We have amended our discussion accordingly.
- I do not agree that Reviews 9,10,11 focus mainly on nucleophile addition reactivity. The description is not correct. Our view was that the reviews did focus "mainly" on the ring-opening reactivity – this description does not exclude the incorporation of other material. However, we agree that this could be misinterpreted as the reviews cited did include some carbene insertion chemistry and are happy to provide a more detailed description. It is also clear that the location of our citations was unhelpful – we wanted to refer to References 9 and 10 as the "recent reviews" which were published prior to us beginning this review. Reference 11 was published in the final stages of our manuscript preparation and provides a thorough overview of BCB chemistry, including some of the recent cycloaddition-type chemistry. This we wanted to highlight separately. This review approaches the topic from a different angle, however (focussing on mechanism and with a section dedicated to biological applications) so believe our review is complementary in approach.
- Include yields on schemes to help readers understand how efficient the method is. We agree that this would be very helpful to the reader and have added yields where they are available to the figures – see response to Reviewer 2.
- The authors wrote: 'the naphthyl group was sensitized....' I would rather argue that the compound was sensitized and not a group. Thank you for this comment. Our aim here was to highlight the importance of the naphthyl group for a successful reaction. However, the carbonyl group and possibly even the BCB C–C orbital likely have an influence on the absorption ability of the molecule. We have rewritten the sentence as follows "The naphthyl group was key to enabling sensitisation by a thioxanthone-based photocatalyst to give 116*"
- The authors wrote 'a vitamin B12-derived Co-porphyrin' This statement is incorrect. Vitamin B12 is based on a corrin ring and not a porphyrin ring. Thank you for pointing this out. We have corrected this mistake.
- Similarly, 'Sm(II) is a potent single-electron-reductant' I would rather put here Sm(II)I2 as a reductant. Thank you for this suggestion. We have made the suggested alteration.
- Figure 5 – box showing Brown's work. In the mechanism, olefin is missing. Thank you for pointing out this error. We have added the styrene reaction partner to the figure.

- The manuscript is written in a too sophisticated, fancy English (unnecessary), just to name a few: a rich tapestry of complex molecules; BCB structure began to reveal itself; have begun to bear fruit. Thank you for this suggestion. We have gone through the manuscript and modified the language at various points. We thank all reviewers for their time taken to review our submission and for making constructive and helpful suggestions for improving the manuscript. We would be delighted if you would consider our manuscript for publication and look forward to hearing from you in due course.

REVIEWERS' COMMENTS:

Reviewer #2 (Remarks to the Author):

The authors have earnestly and robustly responded to my comments and those of the other reviewers; I am satisfied with their responses. The manuscript is much improved, and I would recommend its publication in Communications Chemistry.

However, for the sake of being the comprehensive reference article this review intends to be, there are a small number of recent publications involving BCBs which might be valuable to include.

Some that I have noticed are:

- Molander: 10.1021/jacs.2c11501
- Aggarwal: 10.1002/anie.202217064
- Gilmour: 10.1021/acscatal.2c04511
- Glorius: 10.1021/jacs.2c09248
- Gevorgyan: 10.1021/jacs.2c09045

Reviewer #3 (Remarks to the Author):

The authors made their best in improving the manuscript. In this reviewer opinion it is suitable to be published in Communication Chemistry.

There are a few issues which the authors may consider:

1. They wrote 'In this review, we approach the chemistry from a fresh perspective, and include the most recent developments in this rapidly expanding field.'

The sentence was rewritten, but the authors still make the point that their review is unique but each of the published ones is indeed unique in too some extend. I would not go that far, therefore, that it brings 'fresh perspective' This is to readers to decide.

2. The representation of the radical reactivity is missing on Figure 1.

3. If I am not mistaken Piv stands for pivaloyl not (pizaloyl) as written in the review

4. In the following sentence 'Malins and co-workers investigated the cycloaddition of Triazolinediones (TADs, 103)' Triazolinediones – T should not be a capital letter. Same for Ruthenium in the following sentence 'Glorius and co-workers reported another application of Rhodium-catalysed rearrangements of BCBs, cleaving

both an internal and external C–C bond of the BCB to give linear alkene derivatives'

5. As pointed by one of the reviewers, the outlook was not inspiring and remains that way.

Author Responses to Reviewers:

Reviewer 2 (Changes to manuscript highlighted in green)

- ...for the sake of being the comprehensive reference article this review intends to be, there are a small number of recent publications involving BCBs which might be valuable to include.

Some that I have noticed are:

- Molander: 10.1021/jacs.2c11501
- Aggarwal: 10.1002/anie.202217064
- Gilmour: 10.1021/acscatal.2c04511
- Glorius: 10.1021/jacs.2c09248
- Gevorgyan: 10.1021/jacs.2c09045

After discussion with the editors, we made the decision to highlight these most recent references in the outlook section.

Review 3 (Changes to manuscript highlighted in blue)

- They wrote 'In this review, we approach the chemistry from a fresh perspective, and include the most recent developments in this rapidly expanding field.' The sentence was rewritten, but the authors still make the point that their review is unique but each of the published ones is indeed unique in too some extend. I would not go that far, therefore, that it brings 'fresh perspective 'This is to readers to decide. We have rewritten the sentence as follows: "In this review, we approach the chemistry from the perspective of the different structural classes of compounds that can be prepared using BCBs."
- The representation of the radical reactivity is missing on Figure 1. Thank you for highlighting this – we have added general reaction schemes for reactions with both nucleophilic and electrophilic radicals.
- If I am not mistaken Piv stands for pivaloyl not (pizaloyl) as written in the review. Thank you for pointing out this typo. This has been corrected.
- In the following sentence 'Malins and co-workers investigated the cycloaddition of Triazolinediones (TADs, 103)' Triazolinediones – T should not be a capital letter. Same for Ruthenium in the following sentence 'Glorius and co-workers reported another application of Rhodium-catalysed rearrangements of BCBs, cleaving both an internal and external C–C bond of the BCB to give linear alkene derivatives'. Both of these capitalisations have been removed.
- As pointed by one of the reviewers, the outlook was not inspiring and remains that way. We have made further additions to the outlook as requested.

We thank all reviewers for their time taken to review our submission and for making constructive and helpful suggestions for improving the manuscript.